# Skip-gram word embeddings in hyperbolic space

## Abstract

Embeddings of tree-like graphs in hyperbolic space were recently shown to surpass their Euclidean counterparts in performance by a large margin. Inspired by these results, we present an algorithm for learning word embeddings in hyperbolic space from free text. An objective function based on the hyperbolic distance is derived and included in the skip-gram negative-sampling architecture from word2vec. The hyperbolic word embeddings are then evaluated on word similarity and analogy benchmarks. The results demonstrate the potential of hyperbolic word embeddings, particularly in low dimensions, though without clear superiority over their Euclidean counterparts. We further discuss subtleties in the formulation of the analogy task in curved spaces.

## 1 Introduction

Machine learning algorithms are often based on features in Euclidean space, assuming a flat geometry. However, in many applications there is a more natural representation of the underlying data in terms of a curved manifold. Hyperbolic space is a negatively-curved, non-Euclidean space. It is advantageous for embedding trees as the circumference of a circle grows exponentially with the radius. Learning embeddings in hyperbolic space has recently gained interest (Nickel & Kiela (2017); Chamberlain et al. (2017); Sala et al. (2018)). So far most works on hyperbolic embeddings have dealt with network or tree-like data and focused on link reconstruction or prediction as evaluation measures. However, the seminal paper of Nickel & Kiela (2017) suggested from the outset a similar approach to word embeddings. This paper presents such an algorithm for learning word embeddings in hyperbolic space from free text and investigates if similar performance gains can be observed as for the graph embeddings. A more detailed motivation to support the choice of hyperbolic space is given in section 3.

The contributions of this paper are the proposition of an objective function for skip-gram on the hyperboloid model of hyperbolic space, the derivation of update equations for gradient based optimisation, first experiments on common word embedding evaluation tasks and a discussion of the adaption of the analogy task to manifolds with curvature.

The paper is structured as follows. In section 2, we summarise prior work on word vector representations and recent works on hyperbolic graph embeddings. Section 3 gives a brief discussion of prior work that connects distributional semantics with hierarchical structures in order to motivate the choice of hyperbolic space as a target space for learning embeddings. In section 4, we introduce notations from Riemannian geometry and describe the hyperboloid model of hyperbolic space. Section 5 reviews the skip-gram architecture from word2vec and suggests an objective function for learning word embeddings on the hyperboloid. In section 6, we evaluate the proposed architecture for common word similarity and analogy tasks and compare the results with the standard Euclidean skip-gram algorithm.

## 2 Related work

Learning semantic representations of words has long been a focus of natural language processing research. Early models for vector representations of words included Latent Semantic Indexing (LSI) (Deerwester et al. (1990)), where a word-context matrix is decomposed by singular value decomposition to produce low dimensional embedding vectors. Latent Dirichlet Analysis (LDA), a

probabilistic framework based on topic modeling that also produces word vectors was introduced by Blei et al. (2003). Neural network models for word embeddings have first emerged in the context of language modeling (Bengio et al. (2003); Mnih & Hinton (2008)), where word embeddings are learned as intermediate features of a neural network predicting the next word from a sequence of past words. The word2vec algorithm, introduced in Mikolov et al. (2013), aimed instead to learn word embeddings that would be useful for a broader range of downstream tasks.

The use of hyperbolic geometry for learning embeddings has recently received some attention in the field of graph embeddings. Nickel & Kiela (2017) use the Poincaré ball model of hyperbolic space and an objective function based on the hyperbolic distance to embed the vertices of a tree derived from the WordNet "is-a" relations. They report far superior performance in terms of graph reconstruction and link prediction compared to the same embedding method in a Euclidean space of the same dimension. Chamberlain et al. (2017) use the Euclidean scalar product rescaled by the hyperbolic distance from the origin as a similarity function for an embedding algorithm and report qualitatively better embeddings of different graph datasets compared to Euclidean space. This amounts to pulling back all data points to the tangent space at the origin and then optimising in this tangent space. Sala et al. (2018) present a combinatorial algorithm for embedding graphs in the Poincaré ball that outperforms prior algorithms and parametrises the trade-off between the required numerical precision and the distortion of the resulting embeddings. In a follow-up paper to the Poincaré embeddings, Nickel & Kiela (2018) use the hyperboloid model in Minkowski space to learn graph embeddings and show its benefits for gradient based optimisation. As we work in the same model of hyperbolic space, their derivation of the update equation is largely similar to ours. Finally, one other recent paper deals with learning hyperbolic embeddings for words and sentences from free text. Dhingra et al. (2018) construct a layer on top of a neural network architecture that maps the preceding activations to polar coordinates on the Poincaré disk. For learning word embeddings, a co-occurrence graph is constructed and embeddings are learned using the algorithm from Nickel & Kiela (2017). Their evaluation shows that the resulting hyperbolic embeddings perform better on inferring lexical entailment relations than Euclidean embeddings trained with skip-gram. However, their hyperbolic embeddings show no advantage for standard word similarity tasks. Moreover, in order to compare the similarity of two words, the authors use the cosine similarity, which is inconsistent with the hyperbolic geometry.

## 3 MOTIVATION FOR HYPERBOLIC EMBEDDINGS

As described in the previous section, hyperbolic space has only recently been considered for learning word embeddings whereas there is a line of research on embedding graphs and trees. However, there are a number of works that suggest the connection of distributional embeddings to hierarchical structures. In Fu et al. (2014), word embeddings learned by skip-gram are used to infer hierarchical hypernym–hyponym relations. It can be observed that these relations manifest themselves in the form of an offset vector that is consistent within clusters of similar relationships. Another example for making use of the hierarchical structure in semantics in the context of word embeddings is hierarchical softmax, where the evaluation of a softmax classifier is optimized during training by traversing a tree of binary classifiers Goodman (2001). It was shown in the case of language modelling that using a semantic tree built from word embeddings by hierarchical clustering improves the results compared to a random tree Mnih & Hinton (2008). Finally, the most prominent example that semantic relationships themselves exhibit a hierarchical structure is WordNet (Miller (1995)), representing manually annotated relations between word-senses as a directed graph.

Although skip-gram learns word embeddings from free text, its aim is to reflect the underlying semantics. The commonly used analogy task as well as the above examples support this claim. Furthermore, those examples suggest that an algorithm that captures semantics will also - at least in part - exhibit the hierarchical structure that is present in semantic relationships. Therefore we propose that hyperbolic space is potentially beneficial for learning word embeddings in the same sense than it is natural for embeddings trees and graphs.

## 4 GEOMETRY OF HYPERBOLIC SPACE

The following sections introduce the hyperboloid model of hyperbolic space together with the explicit formulation of some core concepts from Riemannian geometry. For a general introduction to Riemannian manifolds see e.g. Petersen (2006). We identify points in Euclidean or Minkowski space with their position vectors and denote both by lower case letters. Coordinate components are denoted by lower indexes, as in $v_i$. For a non-zero vector $v$ in a normed vector space, $\hat{v}$ denotes its normalisation, i.e. $\hat{v} = \frac{v}{\|v\|}$.

### 4.1 THE HYPERBOLOID MODEL IN MINKOWSKI SPACE

The relationship of the hyperboloid to its ambient space, called Minkowski space, is analogous to that between the sphere and its ambient Euclidean space. For a detailed account of the hyperboloid model, see e.g. Reynolds (1993).

**Definition 4.1.** The $(n + 1)$-dimensional *Minkowski space* $\mathbb{R}^{(n,1)}$ is the real vector space $\mathbb{R}^{n+1}$ endowed with the *Minkowski dot product*:

$$\langle u, v \rangle_M := \sum_{i=0}^{n-1} u_i v_i - u_n v_n, \tag{1}$$

for $u, v \in \mathbb{R}^{(n,1)}$.

The Minkowski dot product is not positive-definite, i.e. there are vectors for which $\langle v, v \rangle_M < 0$. Therefore, Minkowski space is not an inner product space. A common usage of the Minkowski space $\mathbb{R}^{(3,1)}$ is in special relativity, where the first three (Euclidean) dimensions represent space, and the last time. One common model of hyperbolic space is as a subset of Minkowski space in the form of the upper sheet of a two-sheeted hyperboloid.

**Definition 4.2.** The *hyperboloid model* of hyperbolic space is defined by

$$\mathbb{H}^n = \{ x \in \mathbb{R}^{(n,1)} \mid \langle x, x \rangle_M = -1, \, x_n > 0 \}. \tag{2}$$

The tangent space at a point $p \in \mathbb{H}^n$ is denoted by $T_p \mathbb{H}^n$. It is the orthogonal complement of $p$ with respect to the Minkowski dot product:

$$T_p \mathbb{H}^n = \{ x \in \mathbb{R}^{(n,1)} \mid \langle x, p \rangle_M = 0 \}.$$

$\mathbb{H}^n$ is a smooth manifold and can be equipped with a Riemannian metric by the induced scalar product from the ambient Minkowski dot product on the tangent spaces:

$$\text{For } p \in \mathbb{H}^n, \quad v, w \in T_p \mathbb{H}^n, \quad g_p(v, w) := \langle v, w \rangle_M. \tag{3}$$

The magnitude of a vector $v \in T_p \mathbb{H}^n$ can then be defined as

$$\|v\| := \sqrt{g_p(v, v)} = \sqrt{\langle v, v \rangle_M}. \tag{4}$$

The restriction of the Minkowski dot product yields a positive-definite inner product on the tangent spaces of $\mathbb{H}^n$ (despite not being positive-definite itself). This makes $\mathbb{H}^n$ a Riemannian manifold.

### 4.2 OPTIMISATION IN HYPERBOLIC SPACE

Similar to a model in Euclidean space, stochastic gradient descent can be used to find local minima of a differentiable objective function $f : \mathbb{H}^n \to \mathbb{R}$. However, since hyperbolic space is a Riemannian manifold, the gradient of the function at a point $p \in \mathbb{H}^n$ will be an element of the tangent space $T_p \mathbb{H}^n$. Therefore, adding the gradient to the current parameter does not produce a point in $\mathbb{H}^n$, but in the ambient space $\mathbb{R}^{(n,1)}$. There are several approaches to still use additive updates as an approximation. However, Bonnabel (2011) presents Riemannian gradient descent as a way to use the geometric structure in order to make mathematically sound updates. Furthermore, Wilson & Leimeister (2018) illustrate the benefit of using Riemannian gradient descent in hyperbolic space instead of first-order approximations using retractions. The updates use the so-called exponential

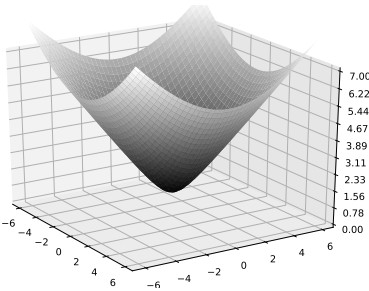

Figure 1: Hyperbolic space as the upper sheet of a hyperboloid in Minkowski space.

map, $\mathrm{Exp}_p$, which maps a tangent vector $v \in T_p\mathbb{H}^n$ to a point on $\mathbb{H}^n$ that is at distance $\|v\|$ from $p$ in the direction of $v$. First, the gradient $\nabla$ of the loss function $f$ with respect to a parameter $p$ is computed. Then the parameter is updated by applying the exponential map to the negative gradient vector scaled by a learning rate $\eta$:

$$p \leftarrow \mathrm{Exp}_p(-\eta \, \nabla f(p)). \tag{5}$$

The paths that are mapped out by the exponential map are called geodesic curves. The geodesics of $\mathbb{H}^n$ are its intersections with two-dimensional planes through the origin. For a point $p \in \mathbb{H}^n$ and an initial direction $v \in T_p\mathbb{H}^n$ the geodesic curve is given by

$$\gamma_{p,v} : \mathbb{R} \to \mathbb{H}^n, \ \gamma_{p,v}(t) = \cosh(\|v\|t) \cdot p + \sinh(\|v\|t) \cdot \hat{v}, \tag{6}$$

where $\hat{v} := \frac{v}{\|v\|}$. The *hyperbolic distance* for two points $p, q \in \mathbb{H}^n$ is computed by

$$d_{\mathbb{H}^n}(p, q) = \mathrm{arccosh}(-\langle p, q \rangle_M). \tag{7}$$

The closed form formulas for geodesics and the hyperbolic distance make the hyperboloid model attractive for formulating optimisation problems in hyperbolic space. In other models the equations take a more complicated form (c.f. the hyperbolic distance and update equations on the Poincaré ball in Nickel & Kiela (2017)).

## 4.3 PARALLEL TRANSPORT ALONG GEODESICS IN $\mathbb{H}^n$

In order to carry out the analogy task on $\mathbb{H}^n$, the translation of vectors in Euclidean space needs to be generalised to curved manifolds. This is achieved by parallel transport along geodesics. Parallel transport provides a way to identify the tangent spaces and move a vector from one tangent space to another along a geodesic curve while preserving angles and length.

**Theorem 4.1.** Let $p \in \mathbb{H}^n$ be a point on the hyperboloid and $v, w \in T_p\mathbb{H}^n$. Let $\gamma : \mathbb{R} \to \mathbb{H}^n$ be the geodesic with $\gamma(0) = p, \gamma'(0) = v$. Then the parallel transport of $w$ along $\gamma$ is given by

$$\varphi_{p,\gamma(t)}(w) = \langle w, \hat{v} \rangle_M \cdot \frac{\gamma'(t)}{\|\gamma'(t)\|} + w - \langle w, \hat{v} \rangle_M \cdot \hat{v}. \tag{8}$$

For a proof sketch see appendix B.2. This can be used to compute the parallel transport of the vector $w \in T_p\mathbb{H}^n$ to a point $q \in \mathbb{H}^n$, by chosing $\gamma$ to be the geodesic connecting $p$ and $q$, and thus $v = \mathrm{Log}_p(q) := \mathrm{Exp}_p^{-1}(q)$. Given $\gamma(t) = \mathrm{Exp}_p(t \cdot v) = \cosh(t\|v\|) \cdot p + \sinh(t\|v\|)\hat{v}$, the derivative is given by $\gamma'(t) = \sinh(t\|v\|) \cdot p \cdot \|v\| + \cosh(t\|v\|) \cdot \hat{v} \cdot \|v\|$. Therefore,

$$\frac{\gamma'(1)}{\|\gamma'(1)\|} = \sinh(\|v\|) \cdot p + \cosh(\|v\|) \cdot \hat{v},$$

since geodesics are unit speed, i.e. $\|\gamma'(t)\| = const. = \|v\|$. This gives

$$\varphi_{p,q}(w) = \langle w, \hat{v} \rangle_M \cdot (\sinh(\|v\|) \cdot p + \cosh(\|v\|) \cdot \hat{v}) + w - \langle w, \hat{v} \rangle_M \cdot \hat{v} \tag{9}$$

that will be used later to transfer the analogy task to hyperbolic space.

## 5 HYPERBOLIC SKIP-GRAM MODEL

### 5.1 WORD2VEC SKIP-GRAM

The skip-gram architecture was first introduced by Mikolov et al. (2013) as one version of the *word2vec* framework. Given a stream of text with words from a fixed vocabulary $\mathcal{V}$, skip-gram training learns a vector representation in Euclidean space for each word. This representation captures word meaning in the sense that words with similar co-occurrence distributions map to nearby vectors. Given a centre word and a context of surrounding words the task in skip-gram learning is to predict each context word from the centre word. One way to efficiently train these embeddings is *negative sampling*, where the embeddings are optimised to identify which of a selection of vocabulary words likely occurred as context words (Mikolov et al. (2013)).

The centre and context words are parametrised as two layers of a neural network architecture. The first layer, representing the centre words, is given by the parameter matrix $\alpha \in \mathbb{R}^{d \times |\mathcal{V}|}$, with $|\mathcal{V}|$ being the number of words in the vocabulary, and $d$ the embedding dimension. Similarly, the output layer is given by $\beta \in \mathbb{R}^{d \times |\mathcal{V}|}$. For both, the columns are indexed by words from the vocabulary $w \in \mathcal{V}$, i.e. $\alpha_w, \beta_w \in \mathbb{R}^d$.

Let $u \in \mathcal{V}$ be the centre word and $w_0 \in \mathcal{V}$ be a context word. Negative sampling training then chooses a number of noise samples $\{w_1, \dots, w_k\}$. The objective function to maximise for this combination of centre and context word is then

$$\mathcal{L}_{u,w_0}(\alpha, \beta) = \prod_{i=0}^{k} P(y_i | w_i, u) = \prod_{i=0}^{k} \sigma((-1)^{1-y_i} \langle \alpha_u, \beta_{w_i} \rangle_{\mathbb{R}^d}), \tag{10}$$

with the labels

$$y_i = \begin{cases} 1 & \text{if } i = 0 \\ 0 & \text{otherwise.} \end{cases}$$

The parameters $\alpha$ and $\gamma$ are optimised using stochastic gradient descent on the negative log likelihood. After training, the vectors of one parameter matrix (in common implementations the input layer, although other publications use both layers, or an aggregate thereof) are the resulting *word embeddings* and can be used as features in downstream tasks.

### 5.2 AN OBJECTIVE FUNCTION FOR SKIP-GRAM TRAINING ON THE HYPERBOLOID

The Euclidean inner product in the skip-gram objective function represents the similarity measure for two word embeddings. Thus, co-occurring words should have a high dot product. Similarly, in hyperbolic space, one can define a similarity by requiring that similar words have a low hyperbolic distance. Since arccosh is monotone, the hyperbolic distance from equation 7 is proportional to the negative Minkowski dot product. This yields an efficient way to represent the similarity on the hyperboloid by just using the Minkowski dot product as similarity function. However, the Minkowski dot product between two points on the hyperboloid is bounded above by $-1$ (reaching the upper bound if and only if the two points are equal). Therefore, when using it as a similarity function in the likelihood function, we apply an additive shift $\theta$ so that neighbouring points indicate a high probability:

$$P(y|w, u) = \sigma\left((-1)^{1-y}(\langle \alpha_u, \beta_w \rangle_M + \theta)\right) \tag{11}$$

$\theta$ is either an additional hyperparameter or could be learned during training. The full loss function for a centre word $u$, context word $w_0$, and negative samples $\{w_1, \dots, w_n\}$ is similar to equation 10:

$$\mathcal{L}_{u,w_0}(\alpha, \beta) = \prod_{i=0}^{k} P(y_i | w_i, u) = \prod_{i=0}^{k} \sigma\left((-1)^{1-y_i}(\langle \alpha_u, \beta_{w_i} \rangle_M + \theta)\right) \tag{12}$$

By using $\langle p, q \rangle_M = -\cosh(d_{\mathbb{H}^n}(p, q))$, the objective function for a positive (i.e. $y = 1$) sample can be evaluated in terms of the hyperbolic distance between two points in $\mathbb{H}^n$. This leads to the function depicted in Figure 2. The choice of the hyperparameter $\theta$ affects the onset of the decay in the activation. This amounts to optimising for a margin between co-occurring words and negative samples.

Since the parameter matrices $\alpha$ and $\beta$ are indexed by the same vocabulary $\mathcal{V}$, they can also be coupled, using only a single layer that represents both the centre and context words.

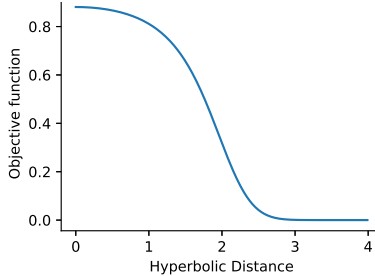

Figure 2: The probability of a sample being positive (with $\theta = 3$).

### 5.3 GEODESIC UPDATE EQUATIONS

To compute the gradient of the objective function $\log \mathcal{L}$, we first compute the gradient $\nabla^{\mathbb{R}^{(n,1)}} \log \mathcal{L}$ of the function extended to the ambient $\mathbb{R}^{(n,1)}$ according to Lemma B.1. Then the Riemannian gradient is the orthogonal projection of this gradient to the tangent space $T_p \mathbb{H}^n$ at the parameter point $p \in \mathbb{H}^n$. For the first layer parameters we get

$$\nabla^{\mathbb{R}^{(n,1)}}_{\alpha_u} \log \mathcal{L}_{u,w_0}(\alpha, \beta) = \sum_{i=0}^{k} \left( y_i - \sigma(\langle \alpha_u, \beta_{w_i} \rangle_M + \theta) \right) \cdot \beta_{w_i}. \tag{13}$$

In a similar fashion, one can compute the gradient for a second layer parameter $\beta_w$. For this, let $\mathcal{S}_u := \{w_0, w_1, \ldots, w_k\}$ be the set of positive and negative samples for the present update step and denote by $\#_{w,\mathcal{S}_u}$ the count of a word $w$ in $\mathcal{S}$. Furthermore, let

$$y(w) = \begin{cases} 1 & \text{if } w = w_0 \\ 0 & \text{if } w \in \{w_1, \ldots, w_k\}. \end{cases}$$

Then the gradient is given by

$$\nabla^{\mathbb{R}^{(n,1)}}_{\beta_w} \log \mathcal{L}_{u,w_0}(\alpha, \beta) = \#_{w,\mathcal{S}_u} \left( y(w) - \sigma(\langle \alpha_u, \beta_{w_i} \rangle_M + \theta) \right) \cdot \alpha_u. \tag{14}$$

Finally both gradients are projected onto the tangent space of $\mathbb{H}^n$. For $p \in \mathbb{H}^n$ and $v \in \mathbb{R}^{(n,1)}$ this is given by

$$\text{proj}_p(v) = v + \langle p, v \rangle_M \cdot p. \tag{15}$$

The resulting projections give the Riemannian gradients on $\mathbb{H}^n$,

$$\nabla^{\mathbb{H}^n}_{\beta_w} \log \mathcal{L}_{u,w_0}(\alpha, \beta) = \text{proj}_{\beta_w} \left( \nabla^{\mathbb{R}^{(n,1)}}_{\beta_w} \log \mathcal{L}_{u,w_0}(\alpha, \beta) \right) \tag{16}$$

$$\nabla^{\mathbb{H}^n}_{\alpha_u} \log \mathcal{L}_{u,w_0}(\alpha, \beta) = \text{proj}_{\alpha_u} \left( \nabla^{\mathbb{R}^{(n,1)}}_{\alpha_u} \log \mathcal{L}_{u,w_0}(\alpha, \beta) \right) \tag{17}$$

that are used for Riemannian stochastic gradient descent according to equation 5.

## 6 EXPERIMENTS

In order to evaluate the quality of the learned embeddings, various common benchmark datasets are available. On the word level, two popular tasks are word similarity and analogy. These will be used here to compare the hyperbolic embeddings with their Euclidean counterparts.

### 6.1 TRAINING SETUP

Word embeddings are trained on a 2013 dump of Wikipedia that has been filtered to contain only pages with at least 20 page views.[1] The raw text has been preprocessed as outlined in appendix A.1.

---

[1] Available at `https://storage.googleapis.com/...`

Table 1: Spearman rank correlation on 3 similarity datasets.

| Dimension/Dataset | Euclidean | | | Hyperbolic | | |
|---|---|---|---|---|---|---|
| | WS-353 | Simlex | MEN | WS-353 | Simlex | MEN |
| 5 | 0.3508 | **0.1622** | 0.4152 | **0.3635** | 0.1460 | **0.4655** |
| 20 | 0.5417 | 0.2291 | 0.6433 | **0.6156** | **0.2554** | **0.6694** |
| 50 | 0.6628 | 0.2738 | **0.7217** | **0.6787** | **0.2784** | 0.7117 |
| 100 | **0.6986** | **0.2923** | **0.7473** | 0.6846 | 0.2832 | 0.7217 |

This results in a corpus of 463k documents with 498 Million words. For learning word embeddings in Euclidean space we use the skip-gram implementation of *fastText*[2], whereas the hyperbolic model has been implemented in C++ based on the fastText code. For the hyperbolic model, the two layers of parameters were identified as this resulted in better performance in informal experiments. The detailed hyperparameters for both models are described in appendix A.2.

## 6.2 WORD SIMILARITY

The word similarity task measures the Spearman rank correlation between word similarity scores (according to the model) and human judgements. We evaluate on three different similarity datasets. The WordSimilarity-353 Test Collection (WS-353) is a relatively small dataset of 353 word pairs, that was introduced in Finkelstein et al. (2001). It covers both similarity, i.e. if words are synonyms, and relatedness, i.e. if they appear in the same context. Simlex-999 (Hill et al. (2015)) consists of 999 pairs aiming at measuring similarity only, not relatedness or association. Finally, the MEN dataset (Bruni et al. (2014)) consists of 3000 word pairs covering both similarity and relatedness. For word embeddings in Euclidean space, the cosine similarity is used as similarity function (Faruqui & Dyer (2014)). We expand this for hyperbolic embeddings by using the Minkowski dot product as similarity function, which is anti-monotone to the hyperbolic distance. For each dimension we report the results of the model with the highest weighted average correlation across the three datasets.

The results are shown in Table 1. The hyperbolic skip-gram embeddings give an improved performance for some combinations and datasets. For the WS-353 and MEN datasets, higher scores can mainly be observed in low dimensions (5, 20), whereas for higher dimensions the Euclidean version is superior by a small margin. The relatively low scores on Simlex-999 suggest that both skip-gram models are better at learning relatedness and association. We point out that our results on the WS-353 dataset surpass the ones achieved in Dhingra et al. (2018), which could potentially be due to their use of the cosine similarity on the Poincaré disk. Overall, we conclude that the proposed method is able to learn sensible embeddings in hyperbolic space and shows potential especially in dimensions that are uncommonly low compared to other algorithms. However, we do not observe the extraordinary performance gains observed for the tree embeddings, where low-dimensional hyperbolic embeddings outperformed Euclidean embeddings by a large margin (Nickel & Kiela (2017)).

## 6.3 WORD ANALOGY

Evaluating word analogy dates back to the seminal word2vec paper (Mikolov et al. (2013)). It relates to the idea that the learned word representations exhibit so called *word vector arithmetic*, i.e. semantic and syntactic relationships present themselves as translations in the word vector space. For example the relationship between a country and its capital would be encoded in their difference vector and is approximately the same for different instances of the relation, e.g. $vec(France) - vec(Paris) \approx vec(Germany) - vec(Berlin)$. Evaluating the extent to which these relations are fulfilled can then serve as a proxy for the quality of the embeddings. The dataset from Mikolov et al. (2013) consists of roughly 20,000 relations in the form $A : B = C : D$, representing "$A$ is to $B$ as $C$ is to $D$". The evaluation measures how often $vec(D)$ is the closest neighbour to $vec(B) - vec(A) + vec(C)$. All vectors are normalised to unit norm before computing the compound vector, and the three query words are removed from the corpus before computing the nearest neighbour.

Using the analogy task for hyperbolic word embeddings needs some adjustment, since $\mathbb{H}^n$ is not a vector space. Rather, the Riemannian structure has to be used to relate the four embeddings of the

---

[2]https://github.com/facebookresearch/fastText

Table 2: Accuracy on the Google word analogy dataset.

| Dimension | 5 | 20 | 50 | 100 |
|---|---|---|---|---|
| Euclidean | 0.0011 | 0.2089 | **0.3866** | **0.5513** |
| Hyperbolic ($Z$) | 0.0020 | **0.2251** | 0.3536 | 0.3636 |
| Hyperbolic ($Z'$) | 0.0008 | 0.0365 | 0.0439 | 0.0437 |

relation. Let $\text{Log}_p$ be the inverse of the exponential map $\text{Exp}_p$. We propose the following procedure as the natural generalisation of the analogy task to curved manifolds such as hyperbolic space:

---

Let $A : B = C : D$ be the relation to be evaluated and identify the associated word embeddings in $\mathbb{H}^n$ with the same symbols. Then

1. Compute $w = \text{Log}_A(B) \in T_A\mathbb{H}^n$.

2. Compute $v = \text{Log}_A(C) \in T_A\mathbb{H}^n$.

3. Parallel transport $w$ along the geodesic connecting $A$ to $C$, resulting in $\varphi_{A,C}(w) \in T_C\mathbb{H}^n$.

4. Calculate the point $Z = \text{Exp}_C(\varphi_{A,C}(w))$.

5. Search for the closest point to $Z$ using the hyperbolic distance.

---

The result of the first step (corresponding to the vector $B - A$ in the Euclidean formulation), is an element of the tangent space $T_A\mathbb{H}^n$ at $A$. In order to "add" this vector to $C$ however, it needs to be moved to the tangent space $T_C\mathbb{H}^n$ using parallel transport along the geodesic connecting $A$ and $C$. Addition in Euclidean space is following a geodesic starting at $C$ in the direction $B - A$. In $\mathbb{H}^n$, this is achieved by following the geodesic along the tangent vector obtained by parallel transport. The resulting point $Z \in \mathbb{H}^n$ can then be used for the usual nearest neighbour search among all words using the hyperbolic distance.

This procedure seems indeed to be the natural generalisation of the analogy task. There is a subtlety, however. The procedure obtains the point $Z$ by beginning at $A$ and proceeding via $C$, and this point $Z$ is then used to search for nearest neighbours. However, in Euclidean space, it would have been equally valid to proceed in the opposite sense, i.e. by beginning at $A$ and proceeding via $B$, and this would also yield a point $Z'$. In Euclidean space, it doesn't matter which of these two alternatives is followed, since the resulting points $Z, Z'$ coincide (indeed, in the Euclidean case the points $A, B, C, Z = Z'$ form a parallelogram). However, in hyperbolic space, or indeed on any manifold of constant non-zero curvature, the two senses of the procedure yield distinct points, i.e. $Z \neq Z'$. Figure 3 depicts the situation in hyperbolic space for a typical choice of points $A, B, C$ and the resultant points $Z, Z'$ on the Poincaré disc model. However, the problem formulation $A : B = C : D$ is not symmetric, as the proposed relation is between $A$ and $B$, not $A$ and $C$. Therefore, we argue that $Log_A(B)$ should be the tangent vector (representing the relation) that gets parallel transported, and not $Log_A(C)$. This amounts to chosing point $Z$ for the nearest neighbour search, not $Z'$. Table 2 shows the performance on the analogy task of the best embeddings from the word similarity task assessment for the two choices. It is evident that using $Z$ performs significantly better. This suggests the correctness of our hypothesis and illustrates that the analogy problem is indeed not symmetric. Interestingly, in the Euclidean setting this does not surface because the four words in question are considered to form a parallelogram and the missing word can be reached along both sides. In comparison with the performance of the Euclidean embeddings, a tendency similar to that observed in the simliartiy task arises. The hyperbolic embeddings outperform the Euclidean embeddings in dimension 20, but are surpassed in higher dimensions. The lowest dimension 5 appers degenerate for both settings.

## 7 CONCLUSIONS AND OUTLOOK

We presented a first attempt at learning word embeddings in hyperbolic space from free text input. The hyperbolic skip-gram model compared favorably to its Euclidean counterpart for some common similarity datasets and the analogy task, especially in low dimensions. We discussed also subtleties inherent in the straight-forward generalisation of the word analogy task to curved manifolds such as

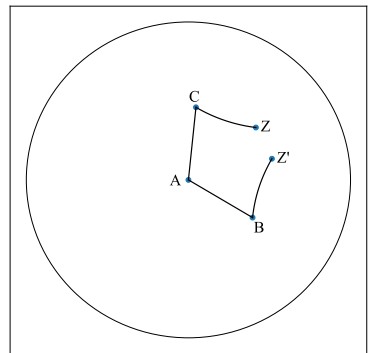

Figure 3: The analogue of the word analogy task in hyperbolic space, depicted using the Poincaré disc model. The curved lines are the geodesic line segments connecting the points, and the opposite sides have the equal hyperbolic length. The generalisation of the word analogy task results in either of two distinct points $Z, Z'$, depending on the choice of going via $B$, or via $C$, having started at $A$.

hyperbolic space and proposed a potential solution. A crucial point for further investigation is the formulation of the objective function. The proposed one is only one possible choice of how to use the hyperbolic structure on top of the skip-gram model. Further experiments might be conducted to potentially increase the performance of hyperbolic word embeddings. Another important direction for future research is the development of the necessary algorithms to use hyperbolic embeddings for downstream tasks. Since many common implementations of classifiers assume Euclidean input data as features, this would require reformulating algorithms so that they can be used in hyperbolic space. In recent work (Ganea et al. (2018), Cho et al. (2018)), hyperbolic versions of various neural network architectures and classifiers were derived. It is hoped this will allow the evaluation of hyperbolic word embeddings on downstream tasks.

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

## A    Implementation details

### A.1    Corpus preprocessing

The preprocessing of the Wikipedia dump consists of lower casing, removing punctuation and retaining the matches of a token pattern that matches words consisting of at least 2 alpha-numeric characters that do not start with a number.

### A.2    Model hyperparameters

For both Euclidean and hyperbolic training we apply a minimum count of 15 to discard infrequent words, use a window size of $\pm 10$ words, 10 negative samples and a subsampling factor of $10^{-5}$. The shift parameter $\theta$ in the hyperbolic skip-gram objective function was set to 3. The distance of the geodesic updates is capped at a maximum of 1 to prevent points moving off of the hyperboloid due to numerical instability. For the hyperbolic model, the two parameter layers are tied and initialised with points sampled from a normal distribution with standard deviation 0.01 around the base point $(0, \ldots, 0, 1)$ of the hyperboloid. For fastText, the default initialisation scheme is used. In both cases, training was run for 3 epochs. For each start learning rate from $\{0.1, 0.05, 0.01, 0.005\}$, the learning rate was decayed linearly to 0 over the full training time. This is one of many common learning rate schemes used for gradient descent in experimental evaluations. However, it does not guarantee convergence. For a detailed account on optimisation on manifolds and conditions on the learning rate that ensure convergence, see Absil et al. (2008).

### A.3    Locking

FastText uses *HogWild* (Niu et al. (2011)) as its optimisation scheme, i.e. multi-threaded stochastic gradient descent without parameter locking. This allows for embedding vectors being concurrently written by different threads. As the Euclidean optimisation is unconstrained, such concurrent writes are unproblematic. In contrast, the hyperbolic optimisation is constrained, since the points must always remain on the hyperboloid, and so concurrent writes to an embedding vector could result in an invalid state. For this reason a locking scheme is used to prevent concurrent access to embedding vectors by separate threads. This scheme locks each parameter vector that is currently in-use by a thread (representing the centre word, or the context word, or a negative sample) so that no other thread can access it. If a thread can not obtain the locks that it needs for a skip-gram learning task, then this task is skipped.

### A.4    Code

The implementation of the hyperbolic skip-gram training and experiments is available online.[3]

## B    Lemmas and proof sketches

### B.1    Gradient in Minkowski space

**Lemma B.1.** For a differentiable function $f : \mathbb{R}^{(n,1)} \to \mathbb{R}$, the gradient is given by

$$\nabla f = \left( \frac{\partial f}{\partial x_0}, \ldots, \frac{\partial f}{\partial x_{n-1}}, -\frac{\partial f}{\partial x_n} \right), \tag{18}$$

where the $\frac{\partial f}{\partial x_i}$ denote partial derivatives according to the Euclidean vector space structure of $\mathbb{R}^{(n,1)}$.

*Proof sketch:* On an embedded (pseudo-)Riemannian submanifold $(M, g)$ of $\mathbb{R}^n$, the Riemannian gradient can be computed by rescaling the Euclidean gradient with the inverse Riemannian metric:

$$\nabla^M = g^{-1} \cdot \nabla^{\mathbb{R}^n}.$$

Minkowski space can be considered a pseudo-Riemannian manifold with metric defined by the Minkowski dot product. The corresponding bilinear form $g$ is the identity matrix with the sign

---

[3]https://github.com/...

flipped in the last component. This gives the formula in terms of the partial derivatives in Lemma B.1.

## B.2 THEOREM 4.1

In this section we show that the formula for parallel transport on $\mathbb{H}^n$ is indeed the parallel transport with respect to the Levi-Civita connection. Since this makes use of intrinsic concepts that are not introduced in the paper, the reader is referred to Petersen (2006) and Robbin & Salamon (2017) for the respective definitions and concepts.

For a smooth curve $\gamma : I \subset \mathbb{R} \to \mathbb{H}^n$, a vector field along $\gamma$ is a smooth map $X : I \to \mathbb{R}^{(n,1)}$ such that $X(t) \in T_{\gamma(t)}\mathbb{H}^n$ for all $t \in I$. The set of all vector fields along a given geodesic $\gamma$ is denoted by Vect$(\gamma)$.

According to Robbin & Salamon (2017), p. 273, for the metric induced on $\mathbb{H}^n$ by the Minkowski dot product, a geodesic $\gamma : \mathbb{R} \to \mathbb{H}^n$ and a vector field $X \in$ Vect$(\gamma)$ , the covariant derivative is given by

$$\nabla X(t) = X'(t) + \langle X'(t), \gamma(t)\rangle_M \cdot \gamma(t) = X'(t) - \langle X(t), \gamma'(t)\rangle_M \cdot \gamma(t). \tag{19}$$

Given an initial tangent vector $v \in T_{\gamma(0)}\mathbb{H}^n$, there is a unique parallel $X \in$ Vect$(\gamma)$ with $X(0) = v$ (Robbin & Salamon (2017), theorem 3.3.4).

In theorem 4.1, the parallel transport $\varphi_{p,\gamma(t)}(w)$ of a vector $w \in T_p\mathbb{H}^n$ along a geodesic $\gamma$ with $\gamma(0) = p$ was claimed to be

$$\varphi_{p,\gamma(t)}(w) = \langle w, \hat{v}\rangle_M \cdot \frac{\gamma'(t)}{\|\gamma'(t)\|} + w - \langle w, \hat{v}\rangle_M \cdot \hat{v}.$$

It can easily be shown that $\varphi_{p,\gamma(t)}(w)$ is smooth as a map $\mathbb{R} \to \mathbb{R}^{(n,1)}$ and is a vector field along $\gamma$, i.e. $\varphi_{p,\gamma(t)}(w) \in T_{\gamma(t)}\mathbb{H}^n$ for all $t$. In order to show that it is also parallel along $\gamma$, we compute

$$\nabla\varphi_{p,\gamma(t)}(w) = \varphi'_{p,\gamma(t)}(w) - \langle\varphi_{p,\gamma(t)}(w), \gamma'(t)\rangle_M \cdot \gamma(t).$$

The first term equates to

$$\varphi'_{p,\gamma(t)}(w) = \langle w, \hat{v}\rangle_M \cdot \frac{\gamma''(t)}{\|\gamma'(t)\|} = \langle w, \hat{v}\rangle_M \frac{\langle\gamma'(t), \gamma'(t)\rangle_M}{\|\gamma'(t)\|} \cdot \gamma(t),$$

since $\gamma$ is a geodesic (see Robbin & Salamon (2017), p. 274).

For the second term we get

$$\langle\varphi_{p,\gamma(t)}(w), \gamma'(t)\rangle_M \cdot \gamma(t) = \langle w, \hat{v}\rangle_M \frac{\langle\gamma'(t), \gamma'(t)\rangle_M}{\|\gamma'(t)\|} \cdot \gamma(t) + \langle w - \langle w, \hat{v}\rangle_M \cdot \hat{v}, \gamma'(t)\rangle_M \cdot \gamma(t).$$

But since $\gamma$ is a geodesic, and the geodesics of $\mathbb{H}^n$ are the intersection of planes through the origin with $\mathbb{H}^n$, we have

$$\gamma'(t) \in \text{span}\{p, v\} \quad \text{and} \quad w - \langle w, \hat{v}\rangle_M \hat{v} \in \text{span}\{p, v\}^\perp$$

Therefore $\langle w - \langle w, \hat{v}\rangle_M \cdot \hat{v}, \gamma'(t)\rangle_M = 0$. Thus, for all $t$,

$$\nabla\varphi_{p,\gamma(t)}(w) = \langle w, \hat{v}\rangle_M \frac{\langle\gamma'(t), \gamma'(t)\rangle_M}{\|\gamma'(t)\|} \cdot \gamma(t) - \langle w, \hat{v}\rangle_M \frac{\langle\gamma'(t), \gamma'(t)\rangle_M}{\|\gamma'(t)\|} \cdot \gamma(t) = 0.$$

$\square$

