# OpenReview forum: "Skip-gram word embeddings in hyperbolic space"
_ICLR.cc/2019/Conference_

### Official Review · AnonReviewer1 · 2018-11-01
**word2vec style embeddings in hyperbolic space**

**Rating:** 6
**Confidence:** 3

**Review:**

This paper presents a technique for embedding words in hyperbolic space, which extends previous non-euclidean methods to non-structured data like free text. The authors provide a new gradient based method for creating the embeddings and then evaluate them on standard word embedding benchmarks. Overall the paper is very well written and well executed. They find that in the low dimensions the approach outperforms standard Euclidean space methods while in higher dimensions this advantage disappears.

The results do not try to claim state of the art on all benchmarks, which I find refreshing and I appreciate the authors candor in giving an honest presentation of their results. Overall, I enjoyed this paper and am eager to see how the authors develop the approach further.

However, along these same lines it would be great to have the authors provide more discussion about the next steps and potential applications for this approach. Is the interest here purely methodological? Are there potential use cases where they believe this approach might be superior to Euclidean approaches? More detail in the discussion and intro about the trajectory of this work would help the reader understand the methodological and application-specific implications.

Pros:
- Clearly written and results are presented in a straightforward manner.
- Extension of analogy reasoning to non-euclidean spaces.

Cons:
- Lack of clear motivation and compelling use case.
- It would be nice to have a visualization of the approach in 2-dimensions. While Figure 3 is instructive for how analogies work in this space, it would be great to visualize an entire dataset. I'm sure that the proposed embeddings would result in  a very different space than euclidean embeddings (as the Poincare embedding paper showed), so it would be great to have at least one visualization of an embedded dataset. Presumably this would play to the strengths of the approach as it excels in lower dimensions.
-  The largest of embedding dimension tested was 100, and it is common to use much larger embeddings of 500-d. Do the trends they observe continue to larger dimensions, e.g. is the performance gap even larger in higher dimensions?

---

> ### Author Response · Authors · 2018-11-26
> **Thank you for your feedback**
>
> Dear Reviewer,
>
> Thank you very much for your feedback and suggestions. We've updated the paper with a section explaining our motivation for learning word embeddings in hyperbolic space. Our main goal was to learn word embeddings in low dimensions, with the future aim of working towards downstream tasks that could then be run with much lower computational complexity. Therefore the evaluation focussed on word level similarity and analogy. The lack of clear superiority compared to Euclidean embeddings is a negative result for our concrete choice of objective function and experimental setup. In that sense the presented approach only provides a first methodological step towards learning word embeddings in hyperbolic space. We did not train in dimensions higher than 100 but conjecture that the trend stays the same, as in prior experiments we observed that for the Euclidean skipgram model the quality of the learned embeddings peaks around dimension 200.

---

### Official Review · AnonReviewer2 · 2018-11-02
**Good theoretical contribution but lack of motivation**

**Rating:** 5
**Confidence:** 3

**Review:**

The paper proposes an algorithm that learns word embeddings in hyperbolic space. It adopts the Skip-Gram objective from Word2Vec on the hyperboloid model and derives the update equations for gradients accordingly.
The authors also propose to compute the word analogy by parallel transport along geodesics on the hyperboloid.

Strength: The paper is well written, both the background geometry and the derived update method are clearly explained. The novelty and theoretical contribution are adequate.

Weakness: My main concern is the lack of motivation for embedding words on the hyperboloid and the choice of evaluation metrics. For Poincare embeddings, the disc area and circle length grow exponentially with their radius, and the distances on the Poincare disk/ball reflect well the hierarchical structure of symbolic data, which make it natural to embed a graph in this space and lead to great evaluation results. The geometric property of the hyperboloid model, however, does not seem to in favor of encoding non-hierarchical semantics of words and the evaluation on word similarity/analogy tasks. The evaluation results in Table 1 and Table 2 show that the hyperbolic embeddings only performs better than the Euclidean embeddings in low dimensions but worse on higher dimensions (>50), while higher dimension embeddings generally encode more semantics and thus are used in downstream tasks. It will be great if the authors could elaborate on the advantages of learning word embeddings in hyperbolic space and evaluate accordingly.

---

> ### Author Response · Authors · 2018-11-26
> **Thank you for your feedback**
>
> Dear Reviewer,
>
> Thank you very much for your feedback and suggestions. We've updated the paper with a section explaining our motivation for learning word embeddings in hyperbolic space. We agree that in order to support this hypothesis it might be interesting to evaluate if the hierarchical nature of language is reflected in the learned embeddings. However, our central goal was to learn good word embeddings in low dimensions. This is why we evaluated using standard word level tasks such as similarity and analogy.

---

### Official Review · AnonReviewer3 · 2018-11-06
**skip-gram + Minkowski distance -> unclear why it should work**

**Rating:** 5
**Confidence:** 3

**Review:**

The present paper aims to apply recent developments in hyperbolic embeddings of graphs to the area of word embeddings.

The paper is relatively clearly written and looks technically correct. The main contribution of the paper is in suggesting the usage of Minkowski dot-product instead of Eucledian dot-product in skip-gram model and derivation of corresponding weight update formulas taking into account the peculiarities of hyperbolic space. The suggested aproach is realtively simple, though requiring adding additional bias parameter, which doesn't look entirely natural for the problem considered. I should note, that all the update equations are relatively straitforward given the results of recent papers on hyperbolic embeddings for graphs. Experimental results show some mild to no improvement over classical skip gram model in word similarity and word analogy problems.

My main concern about the paper is that it is not entirely clear throughout the text why the proposed model should be better than any of the baselines. Currently it lookss like the paper merging 2 ideas without any clear expalanation why they should work well together. I believe that the proposed approach (or similar one) might be useful for practice of natural language processing, but to asses that one would need to base on clear motivation and support this motivation with some examples showing that hyperbolicity indeed helps to capture semantics better (like famous world analogy examples for word2vec).

Pros:
- clearly written
- technically correct
Cons:
- technically straightforward
- not convincing experiments
- unclear, why the approach should work

---

> ### Author Response · Authors · 2018-11-26
> **Thank you for your feedback**
>
> Dear Reviewer,
>
> Thank you very much for your feedback. We've updated the paper with a section explaining our motivation for learning word embeddings in hyperbolic space. As for the concrete results, we are aware that we don't show clear superiority over the Euclidean baseline in all dimensions. In that sense the paper presents a negative result for this concrete choice of loss function and architecture. Methodologically, the paper provides basic work toward training word embeddings on the hyperboloid. Indeed, we note that in the meantime our general approach and the proposed objective function (equation 12) were adopted by other researchers, expanding it to product manifolds and outperforming Euclidean embeddings across dimensions [1].
>
> Regarding the straightforwardness of the update rules in light of prior work, to our knowledge - at the time of writing - all other approaches worked on the Poincaré disk, leading to complicated equations for full Riemannian gradient descent. We are aware that the recent paper by Nickel & Kiela [2] already makes use of the hyperboloid model and effectively derives the gradient updates, in a manner very similar to our presentation. Our work was developed in parallel to theirs and we attributed it in the related works section accordingly.
>
> [1] Anonymous. Learning Mixed-Curvature Representations in Product Spaces. Submitted to ICLR 2019. https://openreview.net/forum?id=HJxeWnCcF7
> [2] Maximilian Nickel, Douwe Kiela. Learning Continuous Hierarchies in the Lorentz Model of Hyperbolic Geometry. ICML 2018

---

### Meta-Review · Area_Chair1 · 2018-12-14
**motivation is unclear**

**Confidence:** 4
**Recommendation:** Reject

**Metareview:**

although the proposed method could be considered an interesting application to recently popular hypobolic space to word embeddings, it is unclear why this needs to be done so. experiments also do not support why or whether the application of hyperbolic space to word embedding is necessary.